# Smartphone-Based Artificial Intelligence for the Detection and Diagnosis of Pediatric Diseases: A Comprehensive Review

**DOI:** 10.3390/bioengineering11060628

**Published:** 2024-06-19

**Authors:** Nicola Principi, Susanna Esposito

**Affiliations:** 1Università degli Studi di Milano, 20122 Milan, Italy; nicola.principi@unimi.it; 2Pediatric Clinic, Department of Medicine and Surgery, University of Parma, 43126 Parma, Italy

**Keywords:** artificial intelligence, acute otitis media, amblyopia, obesity, otitis media with effusion, visual screening

## Abstract

In recent years, the use of smartphones and other wireless technology in medical care has developed rapidly. However, in some cases, especially for pediatric medical problems, the reliability of information accessed by mobile health technology remains debatable. The main aim of this paper is to evaluate the relevance of smartphone applications in the detection and diagnosis of pediatric medical conditions for which the greatest number of applications have been developed. This is the case of smartphone applications developed for the diagnosis of acute otitis media, otitis media with effusion, hearing impairment, obesity, amblyopia, and vision screening. In some cases, the information given by these applications has significantly improved the diagnostic ability of physicians. However, distinguishing between applications that can be effective and those that may lead to mistakes can be very difficult. This highlights the importance of a careful application selection before including smartphone-based artificial intelligence in everyday clinical practice.

## 1. Introduction

With the advancement in mobile communication technologies, mobile health (mHealth), i.e., the use of smartphones and other wireless technology in medical care, has developed rapidly. Many applications have been prepared and made available in the most important search engines at no cost or with a fee. The main aims of most of them are to educate consumers about preventive healthcare services, to favor disease surveillance and treatment support of chronic diseases through a relevant improvement of patient–doctor communication, by promoting careful patient monitoring, avoiding unnecessary use of clinic resources, and maximizing the quality of consultations [1]. In some cases, applications were developed to directly improve medical doctors’ activities by offering diagnostic support in making important clinical decisions through quick access to medical knowledge about diseases, drugs, and the latest clinical discoveries. Some applications were structured to solve diagnostic laboratory problems through sensitive and specific bioanalytical assays, allowing the decentralization of medical specialties and faster data acquisition, storage, and management [2]. Similarly, applications can more rapidly solve diagnostic problems automatically by comparing data found in a single patient with those previously collected and classified by a number of experts in patients with well-defined diseases. Common in this regard are applications that detect lung cancer or strokes based on computer tomography scans, predict the risk of sudden cardiac death or other heart diseases based on electrocardiogram and cardiac magnetic resonance images, classify skin lesions comparing skin images, or identify the risk of diabetic retinopathy in eye images [3]. Finally, applications accessible via a smartphone attached to a simple, cheap instrument can substitute more complex diagnostic machines, which are much more expensive and frequently not easily available. The applications for a diagnosis of the presence of fluid in the middle ear are paradigmatic in this regard [4].

Although most of the presently available applications have been prepared to solve medical problems concerning adults, children were not forgotten. A study has shown that already in 2017, in Google’s Play Store (Google Inc., Mountain View, CA, USA) and Apple’s App Store, 58 applications concerning pediatric topics could be found [5,6]. However, smartphone use by children, parents, and doctors should be carefully evaluated. Some applications can provide questionable information, as most of them are not based on scientific guidelines. Moreover, in some cases, diagnostic conclusions deriving from the comparison of patient data with those in a prepared archive are not validated and can lead to wrong diagnoses [7]. The same seems true for some applications that are associated with simple instruments used to substitute complex and expensive test machines. The main aim of this narrative review is to evaluate the relevance of smartphone applications in the detection and diagnosis of pediatric medical conditions for which the greatest number of applications have been developed. The MEDLINE/PubMed database was searched from 2000 to 30 October 2023 to collect the relevant literature. The search included randomized placebo-controlled trials, controlled clinical trials, double-blind, randomized controlled studies, systematic reviews, and meta-analyses. Only manuscripts including data on medical practice were considered. Abstracts were excluded. The following combinations of keywords were used: “smartphone” AND “artificial intelligence” AND “children” OR “adolescent” OR “paediatric” OR “pediatric”. Ear disorders, eye diseases, and obesity are the topics on which more than 10 studies for each topic were identified and selected for this manuscript.

## 2. Ear Disorders

### 2.1. Acute Otitis Media and Otitis Media with Effusion

Acute otitis media (AOM) is an extremely common disease. By 3 years of age, 60% of children have suffered from ≥1 episode and 24% from ≥3 episodes, making AOM an important medical, social, and economic pediatric problem and one of the most common causes of antibiotic use in children [8,9]. Very common, also, is otitis media with effusion (OME), a condition that follows AOM and is characterized by the presence of fluid in the middle ear (MEF) without signs of infection [10]. About 80% of preschool children experience at least one episode of OME, 30–40% of them have recurrent episodes, and 5–10% have chronic disease lasting for more than one month. OME may result in prolonged decreased hearing, which is associated with communication difficulties, withdrawal, lack of attention, and poor school results [10]. For both AOM and OME, an initial diagnostic approach is made using signs and symptoms of disease and otoscopic findings obtained through a standard otoscope. This is a monocular device that provides only a two-dimensional view of the ear canal and does not allow a detailed evaluation of the tympanic membrane (TM) characteristics and mobility [11,12]. TM evaluation may be difficult due to the narrow ear canal and the presence of cerumen [13]. TM erythema, which is mandatory to confirm inflammation and allow AOM diagnosis, may be missing or present in children who are simply crying or suffering from an upper respiratory infection without OME. The presence of MEF can be evidenced only in the case of TM perforation or AOM with severe TM bulging. Finally, TM modification in the case of OME is not characteristic and does not support OME diagnosis in most cases. Practically, standard otoscopy leads to a certain diagnosis in no more than 50% of OMA and OME cases [14,15]. Diagnostic uncertainty is common, and to avoid risks related to a disease mainly caused by bacteria, some children without AOM are treated with antibiotics and followed as true AOM cases [16]. A similar debatable pharmacologic approach is frequently followed for children with suspected chronic OME, causing an increased risk of selection of resistant bacteria, drug-related adverse events, and increased health expenses. To overcome these limitations, it is suggested to support standard otoscopy with other diagnostic measures. Otomicroscopy, tympanocentesis (TYM), pneumatic otoscopy (PO), tympanometry (TP), and acoustic reflectometer (AR) have been largely studied in this regard. Despite the fact that, in several trials, especially when these diagnostic methods were used in association, they have allowed a significantly reduced misdiagnosis of AOM and OME, their use in outpatients by primary care pediatricians and general practitioners has remained very poor [17]. This is because performing and interpreting results offered by these diagnostic methods require extensive training or extended practice, and, at least in some cases, the instruments are very expensive or too invasive. Otomicroscopy is reserved for ENT specialists as it is an expensive machine with important space limitations. TP necessitates the use of expensive equipment and a referral to an audiologist, as the results remain difficult to evaluate for poorly experienced pediatricians or general practitioners. Moreover, it has the disadvantage of requiring an airtight seal in the ear canal, and the accuracy of the results is thus affected by the cooperation of the child [18,19,20]. AR has similar limitations. TYM is an invasive method that involves the presence of a nurse, is a time-consuming procedure, and is not viewed with high favor by either physicians or parents. Moreover, TYM can be performed only when a bulging TM is documented and, consequently, has no role for OME diagnosis [21]. This explains why it is poorly used, despite being repeatedly suggested by some experts as a simple method to demonstrate MEE and help decide whether and which antibiotics should be prescribed [22]. TYM has not been adopted in primary care practice. PO is probably the best solution for the evaluation of TM characteristics and MEF detection because the instrument is very cheap, and information on TM mobility allows definitive conclusions. However, in routine office practice, PO is difficult to perform, especially in younger children, because of the narrow ear canal and tendency to wriggle [23].

To overcome some of the limitations of standard otoscopy and other diagnostic measures, the use of smartphones can play a very significant role. A first step has been taken with the development of the so-called digital smartphone otoscopes [24,25,26,27,28]. They allow the storage of a great number of the patient’s TM images that can be later reviewed by the physician himself or an ENT specialist. This leads to a deeper analysis of the TM’s characteristics and may resolve doubts and reduce the risk of misdiagnosis [29]. Moreover, smartphone otoscopes can be used to improve the experience of young physicians, reducing the time needed for the acquisition of the knowledge useful for a reliable evaluation of TM characteristics with a standard otoscope. Finally, they can be used at home by parents or guardians, allowing them to image the TM of a sick child and transmit these videos to a physician for remote diagnosis, reducing costs, improving customer satisfaction, and optimizing the physician’s work. Albeit with exceptions [30], studies evaluating the accuracy of these instruments indicate that they can significantly improve the diagnostic accuracy of AOM [31,32] and be useful for the careful control of tympanostomy tube surveillance [33]. Further improvement was obtained when it became possible to compare images collected with video otoscopes with a series of TM images previously collected in patients with well-defined ear diseases applying artificial intelligence (AI). The comparison of the images evidenced in the patient under study with the characteristic findings of certain diagnoses may allow the solution of many doubtful cases, increasing the number of correct diagnoses. On this basis, several studies have developed feature-extraction-based algorithms for the automatic diagnosis of AOM and OME [34,35,36,37]. An example is the algorithm developed by Livingstone and Chau, which obtained 1366 unique otoscopic images from Google Images, from open access repositories, and within otolaryngology clinics [38]. After preprocessing, these images were uploaded to the Google Cloud Vision AutoML platform and annotated with one or more of 14 otologic diagnoses. A consensus set of labels for each otoscopic image was attained, and a multilabel classifier architecture algorithm was trained. The performance of the algorithm was compared to the performance of physicians from different medical specialties. The results were in favor of the algorithm, as the diagnostic accuracy of the algorithm was significantly better (88.7%) than the average physician accuracy (58.9%). More complex is the study by Wu et al., who built an otoscopic image classifier based upon the idea of deep learning and transfer learning using the two most widely used 71-layer and 53-layer convolutional neural networks, named Xception and MobileNet-V2, respectively [39]. The evaluation of the classifier confirmed the expected results, showing sensitivities of 98.4% and 97.6% in the classification of AOM and OME. A more advanced analysis can be obtained with other systems, such as the HearScope system. It is based on two AI systems for classifying otoscopy images. The first screens whether the image actually represents the ear canal and TM. The second network classifies the images according to whether they represent a normal TM, cerumen obstruction, chronic perforations, or abnormal findings (i.e., AOM and OME). A 94% accuracy for classifying images into the four currently supported diagnostic categories was demonstrated [39]. Several other AI-supported otoscopy systems have been developed, confirming that they may assist healthcare workers, trainees, and primary care practitioners with poor otology experience in the identification of ear disease [40]. However, some doubts remain regarding the use of smartphone-based otoscopes by parents or guardians for remote diagnosis of AOM or OME. A study carried out in a small group of children found [41] that the agreement between diagnosis via the current diagnostic standard of PO by a physician and remote diagnosis by CSO using parent-acquired TM videos was very poor, mainly because the images obtained by parents were not suitable for the physician to make the correct diagnosis. Finally, a comparison of the most recently developed systems has shown substantial heterogeneity in performance between studies, making it impossible to evaluate which offers the most reliable results. This seems to indicate that the main obstacle to producing effective algorithms is to base their development on a standardized, robust, comprehensive, and reliable database, which is not easy to achieve [40].

Smartphones can also be used to substitute for standard TP and automatically indicate whether MEF is present. The results obtained with the use of multiple smartphone-based systems have led to interesting results, suggesting that they can significantly simplify TP and offer results substantially similar to those obtained with the standard machine with a significant reduction in economic costs. The system initially described by Chan et al. is the simplest [42]. It uses the speaker of the smartphone to produce audible, frequency-modulated continuous wave chirps that are directed into the ear canal through a paper tunnel prepared with scissors and tape. The sounds reflected by the TM and received by the microphone are analyzed through a logistics learning model capable of classifying the received waveforms and deciding whether they indicate MEF presence [42]. More recently, the same authors developed a more effective instrument that consists of a lightweight and portable attachment to vary air pressure in the ear and measure middle ear function [42]. The smartphone displays a tympanogram and reports peak acoustic admittance in real time. An analysis of its efficacy in MEF detection revealed that the data collected with this smartphone were in good agreement with those produced with standard TP and analyzed by experienced audiologists in 86% of the cases. Finally, recent studies have shown that smartphone-obtained tympanograms can be better analyzed with AI computer vision algorithms developed from various methods to classify otoscopic images. Jin et al. described the diagnostic performance of a hybrid deep learning model for classifying narrow-band tympanometry tracings [43]. A machine learning model was trained and evaluated on 4810 pairs of TP tracings acquired by an audiologist and layperson. It was shown that the sensitivity and specificity of the machine learning model were not substantially different from those acquired by the audiologist. However, it cannot be forgotten that interpretation of the results requires appropriate clinical context, such as the symptoms and time course, and positive results should prompt further clinical evaluation for potential misclassifications. These systems also do not distinguish between different types of middle ear fluid (purulent, serous, or mucoid).

In conclusion, digital smartphone otoscopes and multiple smartphone-based systems appear useful for improving AOM and OME diagnosis. Further studies are needed to evaluate their cost-effectiveness in primary care and hospital settings.

### 2.2. Hearing Impairment

In children, early identification and treatment of hearing impairment are essential to avoid functional and psychosocial problems, mainly limitations in language development, poor learning processes, and social isolation [44]. In most industrialized countries, all neonates are screened with hearing tests, such as the automated otoacoustic emission (AOAE) and the auditory brainstem response (ABR) tests, which can identify most congenital hearing loss cases and allow appropriate intervention [45]. Unfortunately, systematic neonatal screening is not performed in most developing countries. Moreover, some inherited forms of hearing loss do not appear until a child is older. Finally, several factors, such as childhood illness, ear infection, head injury, certain medications, and loud noise are also linked to the development of hearing loss in the first years of life [46]. According to the Centers for Disease Control and Prevention in the USA, up to 5% of preschool children presents with hearing loss [47]. This finding and the relevance of its clinical and social impact led the World Health Organization World Report on Hearing to recommend that all countries implement school screening programs to ensure early detection of ear disease and hearing loss [48].

The cost of hearing screening through traditional audiometric equipment and trained personnel can be prohibitive. To favor the prevention and identification of hearing loss, a large series of simple and cheap methods and instruments for hearing loss screening have been developed. The availability of portable audiometric screening platforms, the access to hearing health services in remote locations, and the applied automated hearing screening test with a smartphone or tablet at home, even in absence of specialized professionals, have significantly improved the identification of hearing impairment [49,50,51,52,53]. Regarding smartphones, at least 44 applications specifically prepared for adults have been developed, and some of them have been found effective in hearing loss screening, particularly in low- and middle-income countries and settings of limited in-person medical care. Most applications measure hearing thresholds as conventional audiometry. They require the use of earphones and, given the huge variety of combinations of earphones and mobile phones, standardized and calibrated software and devices are key to performing reliable hearing tests. Some of these applications can produce an audiogram that can be stored for remote monitoring and management. Unfortunately, most of them have significant problems that limit results availability. In addition to issues with interpretation, audiograms may not be reliable due to the poor calibration of the smartphone speaker or headphones, which does not ensure that the sound level presented to the user is consistent with the sound level intended to be tested. Other applications are simpler but cause even more mistakes, as they indicate only an arbitrary score or a qualitative result as normal hearing or impairment. When the hearing threshold is not measured, tests to evaluate the hearing of the patient in real-world environments are performed, but, also in these cases, mistakes are common. Finally, few applications have been validated, and, among them, uHear, Audcal, Audiogram Mobile, Hearing Test (e-audiologia.pl), Hearing Test Pro (e-audiologia.pl), HearScreen USA, and hearZA are those most frequently downloaded and used [54]. The results on their use, despite being generally good, are not uniform. All these findings justify the conclusions of a recent review that clearly highlights the need of further research and validation efforts to determine whether smartphone-based hearing assessments are feasible and accurate screening tools [54].

In some cases, smartphone applications have similar or even better performances than traditional portable audiometers. An example is the hearScreen application. This was studied in a group of 1070 school-aged children screened twice, once using conventional audiometry and once with the smartphone hearing screening [55]. The two techniques had equivalent sensitivity (75.0%) and specificity (98.5%) in hearing loss detection. Referral rates were, however, lower with the smartphone screening, and it was 12.3% faster than conventional screening. Moreover, in a second study, the same authors showed that this low-cost mobile technology allows minimally trained persons to provide community-based screening comparable to specialized personnel, favoring mass examination in the school [56]. On the contrary, when the uHear application was studied, less satisfactory results were reported [57]. In a group of 38 children aged 4 to 13 years, hearing loss was measured in a quiet environment, and the results were compared with those obtained with traditional audiometry. The results showed that the application was accurate only for hearing loss screening at 2000 Hz in low noise or soundproof rooms, while when tested in a typical environment, uHear showed a lack of accuracy [57]. Limitations were also reported when HearScreen was tested. Manayan et al. showed that this application tested in 208 children had satisfactory sensitivity (85%) and negative predictive value (87%) but low specificity (41%) and positive predictive value (36%) [58]. Moreover, subgroup analysis revealed that it may generate a high proportion of false positives due to the influence of ambient noise on the final results.

To offer an alternative solution for low-cost hearing test equipment and a rough identification of hearing disorders in children, game-based programs (e.g., Sound Scouts and Aud- It) have been developed, which test sound detection and binaural speech processing in any quiet place [59]. These applications use an automatic procedure to determine the hearing threshold of the test subjects. The game concept is that animal sounds are presented, and the test subject should select the correct animal when he hears the sound. In a study involving 1256 children aged 4–13 years testing Sound Scouts, middle/outer ear pathologies were detected in 111 (8.84%). The false positive rate was only 5.01%, showing the potential role of this application in the identification of children with abnormal auditory function [59]. However, in another study on Aud-it that has not undergone peer review testing, it was evidenced that despite being as effective as traditional audiometry in hearing loss identification, the test could be used only in children older than 4 years. Younger children were unable to carry out the test or showed large threshold differences compared to controls tested with traditional audiometry [60]. Moreover, criticisms have been raised by one of Europe’s largest independent research organizations. It has been evidenced that, even if these instruments can be used by non-skilled personnel and allow a selection of children with some ear problems, there are several challenges that need to be resolved before they will work as intended [61].

In conclusion, overall results showed that studies in children are too few, and several problems have to be resolved before smartphone applications can be largely used to screen hearing loss in pediatric age groups.

## 3. Obesity

Studies have shown that childhood obesity (OB) is common worldwide, even in low- and middle-income countries where some decades ago these conditions were very rare, and malnutrition was one of the most common causes of morbidity and mortality [62,63]. The World Health Organization (WHO) has recently estimated that globally at least 38.2 million children under 5 years of age and 340 million children and adolescents aged 5–19 years are overweight or obese [64]. Childhood OB is associated with severe immediate and long-term negative health outcomes that are increasingly presented to health services, along with increased economic costs to individuals and society. Children with these conditions have breathing difficulties and an increased risk of fractures, development of hypertension, type 2 diabetes, and mental health problems [65]. Moreover, as childhood overweight and obesity tend to persist into adulthood, individuals with overweight and obesity during the first year of life frequently develop severe cardiometabolic and psychosocial problems, causing disability and premature death when they become adults [66].

Several factors are associated with OB development. Among them, growing up in an obesogenic environment plays a fundamental role. In recent years, in most countries’ eating habits have changed significantly. Traditional healthy foods have been substituted by ultra-processed, energy-dense, nutrient-poor foods that were preferentially used because they are cheaper, readily available, and strongly sponsored by the food product industry through social media, digital health promotion interventions, digital food marketing, and online food retail [67]. Moreover, the physical activity of children, particularly of those living in industrialized areas, has been significantly reduced both in and out of school due to the loss of walkable green spaces, the rise in motorized transport, and the replacement of outdoor games with sedentary, screen-based leisure activities [68]. To face these problems and promote OB-reducing behaviors, several governments, health organizations, and scientific societies have planned educational programs involving families, schools, and media with the intent to diffuse comprehensive recommendations to increase the use of healthy foods and physical activity in the pediatric population [69]. Unfortunately, the impact of these programs has been lower than expected in most of the countries, as the prevalence of OB has remained essentially unchanged or only marginally reduced, suggesting the need for new types of intervention [70]. Smartphone applications are very attractive in this regard, as they could allow the involvement of larger population groups and provide the opportunity for increased family engagement connected to the health system. Moreover, smartphone applications are frequently free of charge, can be questioned at any time, and have wide distribution and small size. This explains why several smartphone applications potentially useful to improve caregivers’ knowledge of the optimal child diet and nutrition have been prepared and are available via the operating systems’ online stores. In a study considering only applications primarily focused on children <5 years old available free of charge in the Google Play Store at the beginning of 2022, 33 applications were found [71]. The analysis of their characteristics revealed that none of them could be considered potentially effective as a means to improve the quality of diet and nutrition in children. Information regarding nutrient and food requirements according to age was frequently lacking. Moreover, data on diet plans, breastfeeding, complementary feeding, and normal growth were insufficiently discussed. Finally, direct expert consultation was possible only in a limited number of applications, and when it was replaced by a series of answers to supposed frequently asked questions, the clarity and quality of the answers were very poor. Practically, the structure of these applications did not respect the minimum criteria capable of promoting healthy nutrition, e.g., information regarding age-wise nutrients, food requirements, and diet plans, together with videos providing easy recipes and the possibility of a direct expert consultation. Only recently, examples with these characteristics have been developed. Zare et al. developed an application in which the content was based on information in the areas of demographics, assessment, therapeutic recommendations, and application capabilities that, according to a series of international guidelines and expert opinions, are essential to ensure healthy nutrition or to improve the quality of life of children [72]. The application was evaluated for usability in a group of 20 people, including nutritionists and parents. The screen capabilities, the terms and information of the program, learnability, and general features of the application were scored higher than 7.5 out of 9, suggesting the potential for its larger use in the general population. However, as the authors highlighted, the usability and efficacy of the application need confirmation in studies involving a greater number of users chosen from people with lower technological literacy or education levels.

More advanced is the application named MINISTOP 2.0 [73]. It is the most recent evolution of MINISTOP [74], an application developed in Sweden some years ago that was found at least as effective as traditional face-to-face interventions in favoring the use of healthy diets; however, it could not be used outside the country of origin because it was developed only in Swedish. In MINISTOP 2.0, together with the English translation of the text, a series of graphic modifications capable of making the application more explicative for users was introduced. MINISTOP 2.0 contains evidence-based recommendations on diet, physical activity, and screen time, specifically prepared for preschool children. Thirteen topic groups, each released every two weeks, are the base of the application. They include healthy everyday foods, breakfast, snacks, physical activity and screen time, sweets, fruit and vegetables, beverages, snacking, fast food, sleep, meals outside the home, foods as a reward/on special occasions, and dental health. Together with general information for each topic, tips and strategies to reach healthy behavior are reported. Moreover, the application contains a space in which parents can enter what foods and drinks the child has consumed day-by-day, together with information regarding physical activity and screen-watching. Each week, these findings are analyzed, and the results are displayed through graphics and text messages in order to allow parents to evaluate the quality of nutrition and the amount of physical activity in relation to the time spent in front of a screen. The efficacy of the application in conditioning a healthier diet, increased physical activity, and a reduction in screen watching was evaluated in a 6-month study in which 552 parents of 2.5-to-3-year-old children were enrolled [73]. Of them, 277 used MINISTOP 2.0, and 275 simply followed recommendations offered by the Swedish primary child health care system during the routine visit at 2.5–3 years. At the end of the study period, it was shown that children included in the intervention group had lower intakes of sweet and savory treats (6.97 g/day; *p* = 0.001), sweet drinks (−31.52 g/day; *p* < 0.001), and screen time (−7.00 min/day; *p* = 0.012) compared to children in the control group, suggesting potential use of MINISTOP 2.0 for prevention and control of childhood obesity. However, although smartphone-based interventions can facilitate obesity prevention, it is unclear how these relatively novel tools can be implemented in daily practice, and which determinants influence their adoption in healthcare long-term [75]. A study [76] exploring which barriers may limit MINISTOP 2.0 implementation among parents of 4-year-old children revealed that limited knowledge about the application and reservations about how and to what extent parents would use the intervention were the main possible hindrances. Implementation strategies such as educational outreach visits and making the intervention testable among stakeholders could facilitate implementation in the pediatric clinical context.

In conclusion, although smartphone-based AI seems promising for the management of childhood obesity, more research is needed on behavioral change determinants before real-world implementation.

## 4. Eye Diseases

Smartphones are currently used to obtain ocular images and visual measurements of patients’ eyes [77]. In children, one of the eye problems that can be efficiently diagnosed with smartphone applications is amblyopia, a condition that is considered a significant public health concern, with a population prevalence estimated at 2–5% [78]. If not treated early, specifically during the period of visual development between birth and 7 years of age, amblyopia can result in a permanent visual defect or loss of depth perception that can create problems at school, bullying, reduced quality of life, lifelong consequences on future occupation choices, and mental health issues [79]. As initially, amblyopia can be asymptomatic, vision screening to identify amblyopic children is strongly suggested. In the past, this was carried out in the healthcare setting by experienced healthcare professionals or by non-trained professionals in schools, using visual acuity testing in verbal children, or preferential looking tests in preverbal children. Screening is completed by the use of photorefraction with an instrument that identifies the main risk factors for amblyopia, e.g., hyperopia, myopia, astigmatism, strabismus, and anisometropia, and selects children who should be referred for ophthalmological evaluation. Unfortunately, most of these programs were not successful, mainly due to the high cost, limited access to healthcare, and the limited number of qualified screeners [80]. To overcome these limits and make early amblyopia screening possible, screening tools including smartphone applications and other low-cost instruments were developed [81]. The GoCheck Kids software 1.4.1 (Gobiquity Mobile Health, Scottsdale, AZ, USA) is a smartphone photoscreener designed for pediatricians to detect amblyopia risk and can be useful also for the detection of cataract, pupillary abnormalities, and corneal opacities. It has been registered with the U.S. Food and Drug Administration and is intended for vision screening in children from 6 months to 6 years of age [82,83,84]. The system includes a computing device and a smartphone with a photo application, a flash, and a display. Eccentric photorefraction uses a flash source and the phone camera aperture. Photographs are automatically uploaded for manual and automated grading analysis to a secure, compliant study sponsor website using de-identified data. The first version of the software was able to detect hyperopia, myopia, and anisometropia, but not astigmatism, because the refractive error was measured in one meridian only [12,13,14,15,16]. To improve efficacy, a new version with software for capturing images of both eyes simultaneously was developed. Two studies in which this updated smartphone application has been evaluated have shown that its sensitivity and specificity to detect amblyopia risk factors were within the range of traditional instrument-based vision screening technology [85,86]. However, the use of a smartphone took less time, referred more children, and detected a higher number of children with amblyopia than visual acuity testing [87].

These findings seem to indicate that a smartphone photoscreening platform is a promising cost-effective alternative which could assist pediatricians all over the world and minimize obstacles to vision screening and amblyopia detection.

## 5. Conclusions

Smartphones are powerful computing devices with special capabilities, including internet connection and the ability to run various applications. These features, together with the increase in the number of people owning smartphones, have led to the emergence of new health-related interventions known as mHealth. Currently, the use of smartphones is becoming prevalent as an acceptable and easy method for the provision of information, symptoms assessment, and communication with care providers. Moreover, in some situations, smartphone applications have significantly improved the diagnostic ability of physicians, reducing the need for specialist consultation. This is the case in a number of smartphone applications developed for use in pediatrics, such as some of those reported in this review. However, the total number of theoretically available applications for each pediatric sector is very high, and it is not easy, even for experts, to differentiate which can be effective from those that may lead to mistakes. The integration of AI with smartphone technology is revolutionizing pediatric disease diagnosis [88]. These innovations promise to bridge gaps in healthcare accessibility, particularly in underserved areas, by providing accurate, timely, and user-friendly diagnostic tools. Smartphone applications can be useful for both children and physicians, but if their potential usefulness is beyond question, their development and use must still be carefully studied so that they can be relevant in daily practice. Continued research and development in this field are essential to further refine these technologies and expand their capabilities.

## Data Availability

Not applicable.

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
