# Peer review of "Smartphone-Based Artificial Intelligence for the Detection and Diagnosis of Pediatric Diseases: A Comprehensive Review"

_bioengineering, 2024, doi:10.3390/bioengineering11060628_

Round 1

Reviewer 1 Report

Comments and Suggestions for Authors
  1. Abstract section: Please highlight the novel contributions proposed in this manuscript. At first look, the manuscript simply presents a general analysis.
  2. Please avoid the repetition of the same information in both the abstract and introduction sections.
  3. The presentation of section 2.1 must be improved, e.g., the first paragraph is quite long.
  4. All the following sections contain simply long paragraphs quite hard to read and follow. 
  5. The manuscript is missing an analysis of the reported results a way to compare them … or some proof that this manuscript is not simply some summarization of all these published articles.
  6. In general, the manuscript is missing the proposal of some novel contributions.
Comments on the Quality of English Language

Most of the manuscript is hard to read.

Author Response

Re: Thank you for your comments. We revised the manuscript accordingly.

Abstract section: Please highlight the novel contributions proposed in this manuscript. At first look, the manuscript simply presents a general analysis.

Re: Revised (p. 1).

Please avoid the repetition of the same information in both the abstract and introduction sections.

Re: Rephrased (p. 1).

The presentation of section 2.1 must be improved, e.g., the first paragraph is quite long.

Re: Revised (pp. 2-3).

All the following sections contain simply long paragraphs quite hard to read and follow.

Re: Revised.

The manuscript is missing an analysis of the reported results a way to compare them … or some proof that this manuscript is not simply some summarization of all these published articles. In general, the manuscript is missing the proposal of some novel contributions.

Re: We improved the Conclusions according to your suggestions.

Reviewer 2 Report

Comments and Suggestions for Authors

- In general, the idea of the paper is good, however, much is left to be desired. For example, behavioral analysis of pediatrics is missing. Also, the paper is poorly organized with no graphics nor an in-depth analysis. I would have liked to see more about datasets, classification, object detection, audio and video analysis using AI methods, etc. Also, it would be great to include research gaps.

- The similarity ithenticate report shows great similarity with other sources, please rewrite. 

- Lines 9-13, rephrase and make more crisp and succinct. Also, the start of the sentence at line 13, should have "The".

- Line 13, writing that the aim of the paper to discuss, short sells the article, as a discussion does not quality for publication by itself.

- The title should include that this works is a review. 

- Lines 17-19, the sentence is a mess. rephrase.

- Lines 23-25, these do not add anything new and these statements are now well-known facts.

- Very long sentences, e.g., line 55-59, one sentence is five lines. 

- Some devices, e.g., CellScope Oto have been discontinued.   

- Works integrating smartphones, AI, and medical applications can be discussed and cited, see Alafeef, M., Fraiwan, M. Smartphone-based respiratory rate estimation using photoplethysmographic imaging and discrete wavelet transform. J Ambient Intell Human Comput 11, 693–703 (2020). https://doi.org/10.1007/s12652-019-01339-6

- The table of abbreviations is missing but maybe required by the journal template. 

Comments on the Quality of English Language

See the comments to the authors. In general, long sentences with grammatical mistakes all over the manuscript.

Author Response

- In general, the idea of the paper is good, however, much is left to be desired. For example, behavioral analysis of pediatrics is missing. Also, the paper is poorly organized with no graphics nor an in-depth analysis. I would have liked to see more about datasets, classification, object detection, audio and video analysis using AI methods, etc. Also, it would be great to include research gaps.

Re: Thank you for your suggestions. We revised the manuscript according to your comments.

- The similarity ithenticate report shows great similarity with other sources, please rewrite.

Re: Revised (p. 1).

- Lines 9-13, rephrase and make more crisp and succinct. Also, the start of the sentence at line 13, should have "The".

Re: Rephrase as suggested (p. 1).

- Line 13, writing that the aim of the paper to discuss, short sells the article, as a discussion does not quality for publication by itself.

Re: Done (pp. 1-2).

- The title should include that this works is a review.

Re: Revised (p. 1).

- Lines 17-19, the sentence is a mess. rephrase.

Re: Clarified (p. 1).

- Lines 23-25, these do not add anything new and these statements are now well-known facts.

Re: Revised (p. 1).

- Very long sentences, e.g., line 55-59, one sentence is five lines.

Re: Rephrased (p. 2).

- Some devices, e.g., CellScope Oto have been discontinued.   

Re: Revised (p. 4).

- Works integrating smartphones, AI, and medical applications can be discussed and cited, see Alafeef, M., Fraiwan, M. Smartphone-based respiratory rate estimation using photoplethysmographic imaging and discrete wavelet transform. J Ambient Intell Human Comput 11, 693–703 (2020). https://doi.org/10.1007/s12652-019-01339-6

Re: Added (pp. 10 and 15).

- The table of abbreviations is missing but maybe required by the journal template.

Re: It is not required by the journal’s instructions for authors.

Reviewer 3 Report

Comments and Suggestions for Authors

The review article presents a summary of the smartphone applications that utilize artificial intelligence for the detection and diagnosis of pediatric medical conditions. It mainly focuses on those medical conditions for which the greatest number of smartphone applications have been developed.

The review is well written and covers relevant topics with appropriate references. There has been an increasing use of smartphones for remote monitoring and telemedicine applications. With the advent of artificial intelligence, smartphone applications are getting better at detecting and diagnosing medical conditions. Therefore, this review is relevant.

Here are some specific comments/suggestions for the authors:

1. Lines 14-16 : "The main aim of this paper is to ........ in the diagnosis and treatment of those pediatric medical conditions for which the greatest number of applications has been developed." The smartphone applications discussed in the manuscript are being used for the detection and diagnosis of medical conditions and not for the treatment of medical conditions. Consider rewording accordingly.

2. Lines 75-78: "Main aim of this narrative review is to........in the diagnosis and treatment of those pediatric medical conditions for which ...... obesity." Same suggestion as in #1.

3. Lines 14-16 and Lines 75-78: The authors use the phrase, "for which the greatest number of applications has been developed". How did the authors determine the pediatric medical conditions for which the greatest number of applications have been developed? It would be beneficial for the reader if this information were elaborated in the manuscript.

4. Consider including tables under each section (sections 2-4) summarizing the smartphone applications and their applications.

Comments on the Quality of English Language

The author need to check the entire manuscript for English grammar as there are multiple instances of errors. Some examples are: Lines 44-47, Lines 73-75.

Author Response

The review article presents a summary of the smartphone applications that utilize artificial intelligence for the detection and diagnosis of pediatric medical conditions. It mainly focuses on those medical conditions for which the greatest number of smartphone applications have been developed.

The review is well written and covers relevant topics with appropriate references. There has been an increasing use of smartphones for remote monitoring and telemedicine applications. With the advent of artificial intelligence, smartphone applications are getting better at detecting and diagnosing medical conditions. Therefore, this review is relevant.

Re: Thank you for your positive comments. We revised the text according to your recommendations.

Here are some specific comments/suggestions for the authors:

  1. Lines 14-16 : "The main aim of this paper is to ........ in the diagnosis and treatment of those pediatric medical conditions for which the greatest number of applications has been developed." The smartphone applications discussed in the manuscript are being used for the detection and diagnosis of medical conditions and not for the treatment of medical conditions. Consider rewording accordingly.

Re: Revised as suggested (p. 1).

  1. Lines 75-78: "Main aim of this narrative review is to........in the diagnosis and treatment of those pediatric medical conditions for which ...... obesity." Same suggestion as in #1.

Re: Revised as suggested (p. 1).

  1. Lines 14-16 and Lines 75-78: The authors use the phrase, "for which the greatest number of applications has been developed". How did the authors determine the pediatric medical conditions for which the greatest number of applications have been developed? It would be beneficial for the reader if this information were elaborated in the manuscript.

Re: Methodology is reported in details (p. 2).

  1. Consider including tables under each section (sections 2-4) summarizing the smartphone applications and their applications.

Re: We added a brief description of the conclusions after each section as requested by another reviewer (pp. 5, 6, 8 and 9).

Comments on the Quality of English Language

The author need to check the entire manuscript for English grammar as there are multiple instances of errors. Some examples are: Lines 44-47, Lines 73-75.

Re: The text has been revised by an English mothertongue.

Round 2

Reviewer 1 Report

Comments and Suggestions for Authors

  1. The manuscript received a quite minor revision.  
  2. All sections contain long paragraphs quite hard to read and follow.
  3. There is no analysis of the reported results; a (comprehensive) review paper must also provide a way to compare all these articles not simply provide a summary.

In my opinion, the manuscript does not provide any additional value. The authors fail to provide arguments on why would someone read and more important cite such work. A lot more work is required before accepting such a manuscript.

Comments on the Quality of English Language

The manuscript is hard to read.

Author Response

    The manuscript received a quite minor revision.  All sections contain long paragraphs quite hard to read and follow. There is no analysis of the reported results; a (comprehensive) review paper must also provide a way to compare all these articles not simply provide a summary.

Re: Thank you for your comments. We further revised the text according to your recommendations (pp. 5, 6, 8 and 9).

In my opinion, the manuscript does not provide any additional value. The authors fail to provide arguments on why would someone read and more important cite such work. A lot more work is required before accepting such a manuscript.

Re: We hope that you could appreciate and accept this revised version as the other two reviewers.

Reviewer 2 Report

Comments and Suggestions for Authors

The authors addressed my comments.

Author Response

Thank you very much for the approval of our manuscript. We further revised it according to the reviewers' suggestions.